# Structural basis of the mycobacterial stress-response RNA polymerase auto-inhibition via oligomerization

Zakia Morichaud [1], Stefano Trapani [2], Rishi K. Vishwakarma[1,4], Laurent Chaloin[1], Corinne Lionne [2], Joséphine Lai-Kee-Him[2], Patrick Bron [2] ✉ & Konstantin Brodolin [1,3] ✉

Self-assembly of macromolecules into higher-order symmetric structures is fundamental for the regulation of biological processes. Higher-order symmetric structure self-assembly by the gene expression machinery, such as bacterial DNA-dependent RNA polymerase (RNAP), has never been reported before. Here, we show that the stress-response $\sigma^B$ factor from the human pathogen, *Mycobacterium tuberculosis*, induces the RNAP holoenzyme oligomerization into a supramolecular complex composed of eight RNAP units. Cryo-electron microscopy revealed a pseudo-symmetric structure of the RNAP octamer in which RNAP protomers are captured in an auto-inhibited state and display an open-clamp conformation. The structure shows that $\sigma^B$ is sequestered by the RNAP flap and clamp domains. The transcriptional activator RbpA prevented octamer formation by promoting the initiation-competent RNAP conformation. Our results reveal that a non-conserved region of $\sigma$ is an allosteric controller of transcription initiation and demonstrate how basal transcription factors can regulate gene expression by modulating the RNAP holoenzyme assembly and hibernation.

Transcription is the first and most highly regulated step in gene expression. In bacteria, transcription is carried out by the RNA polymerase (RNAP) core (E) composed of five subunits ($\alpha_2\beta\beta'\omega$)[1]. To initiate transcription, the core associates with the promoter-specific initiation factor $\sigma$ (the $\sigma$ subunit) to form the RNAP holoenzyme (E$\sigma$)[2]. Upon holoenzyme assembly, the $\sigma$ subunit domain 2 ($\sigma2$) binds to the $\beta'$ subunit coiled-coil region ($\beta'$-CC) at the RNAP clamp domain while domain 4 ($\sigma4$) binds to the $\beta$ subunit flap ($\beta$ flap). The $\sigma$ subunit region 3.2, which connects domains $\sigma2$ and $\sigma4$ (Fig. 1a), enters deeply into the RNAP active site cleft and RNA exit channel[3–5]. In the holoenzyme, $\sigma$ adopts a conformation optimal for recognition of −10 and −35 promoter elements by $\sigma2$ and $\sigma4$, respectively[6,7]. However, the molecular mechanisms of $\sigma$ loading onto RNAP remain obscure. To initiate RNA synthesis, RNAP melts ~13 bp of promoter DNA and forms the transcriptionally competent open promoter complex (RPo). During transcription initiation, the RNAP clamp successively adopts different states, from open (free RNAP) to closed (RPo and transcription elongation complex)[8–13]. The open-clamp state allows the DNA template entry into the active site, and the closed state is required to hold the DNA template in the active site. Thus, the clamp conformational dynamics drive promoter recognition and melting during transcription initiation[10,12]. The $\sigma2$ and $\sigma4$ domains impose restraints on the clamp and $\beta$ flap relative movements and consequently, should affect the basic RNAP functions in a promoter-dependent manner. Lineage-specific $\sigma$ factors create gene regulatory networks that ensure the rapid adaptation of bacteria to environmental stress and allow pathogenic bacteria to tolerate antibiotic treatment[14,15]. For instance, the dormant form of *M. tuberculosis* (Mtb), the origin of latent

[1]Institut de Recherche en Infectiologie de Montpellier, Univ Montpellier, CNRS, Montpellier 34293, France. [2]Centre de Biologie Structurale, Univ Montpellier, CNRS, INSERM, Montpellier, France. [3]INSERM, Montpellier, France. [4]Present address: Department of Biochemistry & Molecular Biology, The Pennsylvania State University, University Park, PA 16802, USA. ✉e-mail: patrick.bron@cbs.cnrs.fr; konstantin.brodolin@inserm.fr

tuberculosis, can persist in tissues for decades[16]. Most Mtb genes are transcribed by RNAP harboring the principal $\sigma^A$ subunit and the principal-like $\sigma^B$ subunit[15] that belong to group I and group II, respectively, and share almost identical promoter binding regions[17]. It is thought that the $\sigma^B$ subunit is responsible for gene transcription during starvation and stress[18–20]. However, recent findings showed that the mycobacterial $\sigma^A$ and $\sigma^B$ subunits are present at similar levels and co-transcribe essential genes during exponential growth[15]. Two general transcription factors, CarD and RbpA, regulate Mtb RNAP activity in the growth phase and in a gene-specific manner[21–23]. Unlike $E\sigma^A$, the activity of which displays loose dependence on RbpA, $E\sigma^B$ is deficient in promoter-dependent transcription initiation in the absence of RbpA[17,24]. Single-molecule Förster resonance energy transfer (smFRET) analysis showed that RbpA induces the $\sigma^B$ conformational change required for the correct assembly of active $E\sigma^{B6}$. Several solution structures of the Mtb $E\sigma^A$ holoenzyme and its complexes with CarD, RbpA, and promoter DNA have been solved by cryogenic electron microscopy (cryo-EM)[25–27], and have provided the structural basis for understanding $\sigma^A$-dependent transcription initiation. However, the lack of $E\sigma^B$ structure does not allow dissecting the specific roles of the $\sigma^A$ and $\sigma^B$ subunits in gene regulation. Here, we use single-particle cryo-EM to determine the structural basis of the intrinsically limited $E\sigma^B$ transcriptional activity. We find that after $\sigma^B$ association with the RNAP core, the $E\sigma^B$ holoenzyme remains trapped in an immature conformation in which the $\sigma^B$ C-terminus is unloaded from the RNA exit channel. The immature $E\sigma^B$, deficient in promoter recognition, self-assembles into a 3.2 MDa, octamer the size of which exceeds that of the bacterial ribosome. Thus, $\sigma^B$ acts as a bona fide RNAP-hibernation factor that can repress transcription.

## Results

### Cryo-EM structure of the *M. tuberculosis* $E\sigma^B$ octamer

For the structural analysis, we first assembled the $E\sigma^B$ holoenzyme from the separately purified $\sigma^B$ subunit and Mtb RNAP core (Fig. 1b). Analysis of the cryo-EM images revealed two particle populations: RNAP monomers and O-shaped RNAP oligomers (Fig. 1c). We determined the RNAP monomer structure at a nominal resolution of 4.1 Å (Supplementary Figs 1, 2 and Supplementary Table 1). In the cryo-EM map of the monomer, the $\sigma^B$ subunit density was absent. Thus, we concluded that monomer particles comprised mainly the RNAP core. Overall, the structure was similar to the published structure of the *Mycobacterium smegmatis* RNAP core (conformation 2)[28] with a ~5 Å bigger distance between the RNAP β lobe 1 (β-L275) and β′ clamp (β′-R214) domains. We then refined the RNAP oligomer cryo-EM map without imposing symmetry at a nominal resolution of 6.13 Å (Fig. 1d and Supplementary Fig. 3, C1 -map). Analysis of the $C_1$-map showed that the RNAP-oligomer was formed by eight $E\sigma^B$ units assembled into an octamer that exhibited a pseudo dihedral ($D_4$) symmetry (Fig. 1e). For further referencing, we named the RNAP protomers in the $C_1$-map R1 to R8 (Fig. 1d, e). Overall, the $C_1$ -map was non-uniform: the density of the RNAP subunits was well-defined in protomers R1, R2, R5, and R8, (volume of protomers >400 nm³, Fig. 1f) whereas it was incomplete in protomers R3, R4, R6 and R7 (volume of protomers <400 nm³, Fig. 1f). The heterogeneous ab-initio reconstruction with five classes showed that oligomer particles were a mixture of RNAP tetramers that comprised protomers R1, R2, R5 and R8 (~20%) and of RNAP octamers (~80%) (Supplementary Figs 3 and 5a). Refinement of the octamer with the applied $D_4$ symmetry resulted in an improved $D_4$-map at a nominal resolution of 4.39 Å (Supplementary Figs 3, 4a and Supplementary Table 1).

The $E\sigma^B$ octamer structure was formed by two stacked rings, related by twofold symmetry. Each ring was made of four $E\sigma^B$ units, related by fourfold symmetry, in head-to-tail orientation (Fig. 1e, g). The head included the RNAP pincers, formed by β and β′ subunits (Fig. 1g, in cyan and pink respectively), and the tail comprised two α

subunits (Fig. 1g, in orange/yellow). The ring diameter was ~275 Å and the stack height was ~248 Å. The junction between rings was formed by the $\sigma^B$ subunit domain 2 ($\sigma_2^B$, aa 23-158) that interacted with the β′ clamp (aa 1–413) and β flap (aa 808–832) (Fig. 1g, boxed region). The total buried surface area between $\sigma_2^B$ in the R1 protomer and all other chains in the R5 and R8 promoters was ~1200 A², which is similar to that between $\sigma_2^B$ and the β′ subunit in the RNAP holoenzyme (~1300 A²). The β clamp contacts were formed by the Actinobacteria-specific insertion in the β′ subunit (β′i1, aa 141–230) and invariant residues of the β′ $Zn^{2+}$ binding domain (β′ZBD, aa 60-81). Local resolution calculations showed that the σ2-β′ clamp module was determined at a resolution between 3.6 and 5.5 Å while the remaining RNAP parts were determined at a resolution between 5 and 10 Å (Supplementary Fig. 4a). Thus we concluded that the σ2/β′ clamp junction between rings forms a rigid scaffold to hold the whole complex together. The 3D variability analysis[29] demonstrated that the RNAP protomers underwent concerted movements relative to the scaffold, which explained the low resolution in the peripheral zones of the $E\sigma^B$ octamer (Supplementary Fig. 5 and Supplementary Movie 1). The N-terminal part of the $\sigma^B$ subunit, (i.e., domain $\sigma_2^B$ (Fig. 1a) that comprises residues 17 to 158) was well resolved. Little variation in the $\sigma_2^B$ density volume between the eight protomers in $C_1$ -map suggested that majority of the octamer molecules contain eight copies of the $\sigma^B$ subunit (Supplementary Fig. 5d, e). The C-terminal part of $\sigma^B$ (i.e., the σ3 and σ4 domains) was poorly delimited, possibly due to its high mobility.

### Cryo-EM structure of the $E\sigma^B$ protomers

To better characterize the $E\sigma^B$ protomer structure, we performed local refinement of the octamer $C_1$ -map with masked R1 and R5 protomers which displayed better defined density for $\sigma^B$ (Supplementary Fig. 3). We determined the cryo-EM maps of both protomers at a nominal resolution of 3.8 Å (Fig. 2a, Supplementary Fig. 4b and Supplementary Table 1). Although the cryo-EM map of R1 displayed better resolved electron density for $\sigma_2^B$ than that of R5, in both maps the $\sigma^B$ C-terminal domain density was absent. Local resolution calculations of the R1 $E\sigma^B$ map showed that the central part of the Mtb RNAP core was determined at a resolution between 3.4 and 3.7 Å and displayed well-defined structural elements (Fig. 2, Supplementary Figs 4b and 6). Domain $\sigma_2^B$ was determined at resolution between 3.8 and 5 Å. The mobile/flexible peripheral domains: β′jaw, β lobes, and β′i1 were determined at a resolution of 4–6 Å.

To improve the $E\sigma^B$ protomer resolution by including structural information from all eight protomers that constitute $E\sigma^B$ octamer, we tested an alternative reconstruction method based on the symmetry-expansion procedure[30,31]. Briefly, each particle image used to refine the $E\sigma^B$ octamer map was replicated and 3D-rotated according to the $D_4$ point group symmetry. Then, we performed alternate cycles of asymmetric 3D classification and local refinement focused on a single $E\sigma^B$ protomer. Two selected 3D classes with slightly different (11°) aperture of the β′ clamp were refined at 3.9 and 4.19 Å resolution respectively (Supplementary Fig. 7). However, we did not observe any improvement in the resolution of the $\sigma^B$ subunit density map.

### Cryo-EM structure of the $E\sigma^B$ dimer

To better determine the structure of the interactions holding together the $E\sigma^B$ protomers, we performed local refinement of the octamer $C_1$-map with the masked R1-R5 $E\sigma^B$ dimer that displayed better-delimited cryo-EM density for the $\sigma^B$ C-terminal segment ($\sigma^B$ CTS) (Fig. 2b and Supplementary Fig. 3). We determined the $E\sigma^B$ dimer structure at a nominal resolution of 4.36 Å (Supplementary Fig. 4c and Supplementary Table 1). In the RNAP dimer, the $\sigma^B$ C-terminal domains of the two $E\sigma^B$ protomers were stacked together and comprised three disconnected densities, determined at a resolution between 5.5 and 8 Å (Fig. 2b). To improve the $\sigma^B$ C-terminal segment resolution, we performed a 3D classification using the R1-R5 $E\sigma^B$ dimer map as reference.

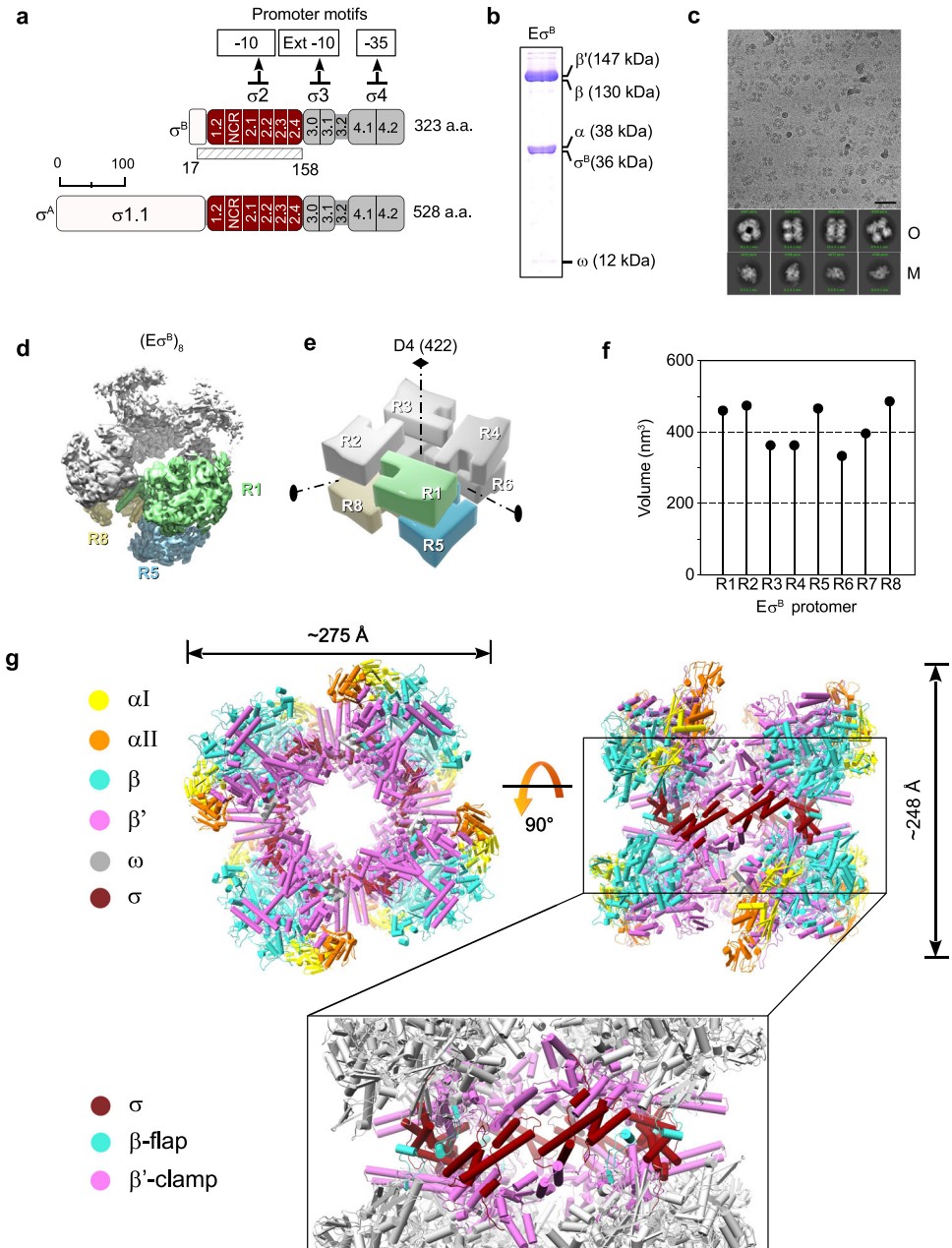

**Fig. 1 | Cryo-EM structure of the *M. tuberculosis* Eσ$^B$ octamer. a** Schematic representation of the *M. tuberculosis* σ$^A$ and σ$^B$ subunits with the structural domains σ1.1 in white, σ2 in dark red, σ3 and σ4 in gray. The subregions inside the σ domains are numbered. NCR, non-conserved region. The solved part of σ$^B$ (residues 17–158) is indicated by a hatched rectangle. **b** 14% SDS-PAGE of the Eσ$^B$ sample for cryo-EM. **c** Representative cryo-EM image of the Eσ$^B$ sample. Experiment was repeated independently four times. Scale bar = 50 nm. Bottom, representative 2D class averages of monomers (M) and oligomers (O). **d** Cryo-EM map of the octamer (Eσ$^B$)$_8$ refined without imposing symmetry (C$_1$-map). RNAP protomers with well-defined density are in light green (protomer R1), sky blue (protomer R5), and khaki (protomer R8). The other protomers are in gray. **e**. 3D-model of the D$_4$ symmetric octamer with the protomers numbered R1 to R8 and the symmetry axes indicated. Color codes are as in **d**. **f** Scatter chart shows volumes of the individual protomers in C$_1$-map calculated in UCSF Chimera. **g** Molecular model of the octamer (Eσ$^B$)$_8$. Views from the top (protomers R1, R2, R3, R4) and from the side (protomers R2, R3, R7, R8) with the RNAP subunits color-coded as indicated on the left: yellow and orange α, cyan β, magenta β′, dark red σ$^B$. The boxed region shows the junction between the top and bottom rings of RNAP tetramers (Eσ$^B$)$_4$. Domains holding the RNAP protomers together: β flap (aa 808–832, cyan), β′ clamp (aa 1–413, magenta) and $\sigma_2^B$ (dark red). The other regions are in gray. Source data are provided as a Source Data file.

This gave two classes: class 1 (58% of particles; nominal resolution: 4.38 Å) with well-defined density of the σ$^B$ C-terminal segment (resolution between 6.5 and 8 Å), and class 2 (42% of particles; nominal resolution: 6.75 Å) that lacked density of the σ$^B$ C-terminal segment (Supplementary Figs 3, 8 and Supplementary Table 1). The relative orientation and conformation of the RNAP protomers in class 1 were the same as in the reference Eσ$^B$ dimer model. Conversely, the relative orientation of RNAP protomers in class 2 was different. In agreement

with the 3D variability analysis of the RNAP octamer, the β lobes and β flaps of the RNAP protomers from class 1 were positioned closer to each other than those in class 2, and thus restrained the movements of the σ$^B$ C-terminal domains (Fig. 2d, Supplementary Fig. 5c and Supplementary Movie 1). Structure analysis showed that the central density of the σ$^B$ C-terminal segment dimer was formed by two copies of tangled polypeptide chains (a1 in R1 and a2 in R5), stacked between the β lobes and β flaps of the neighboring RNAPs (Fig. 2b and

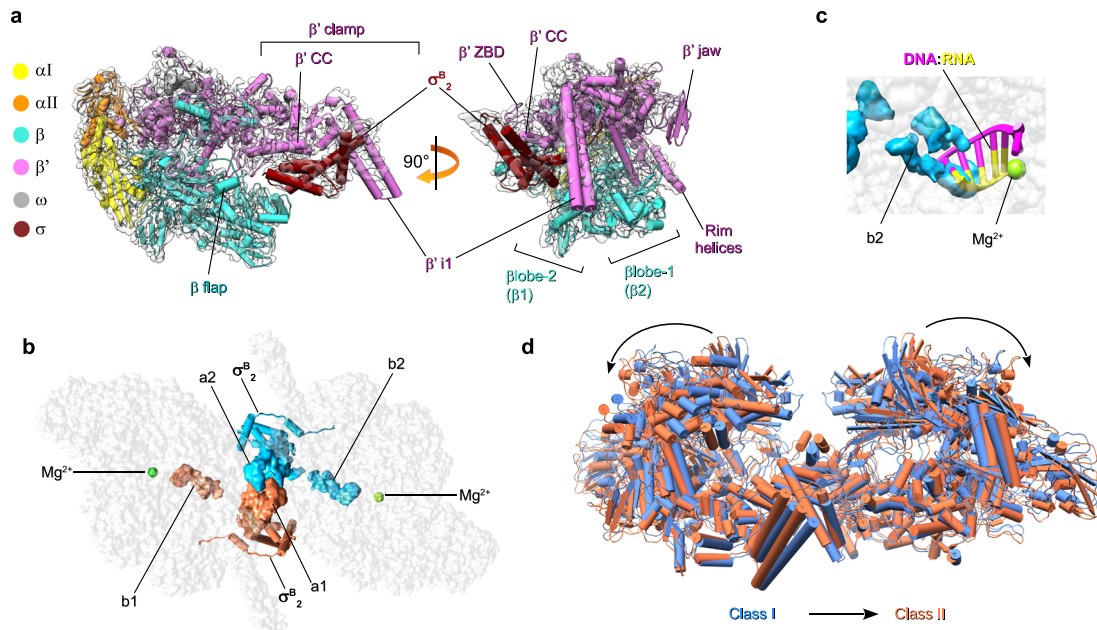

**Fig. 2 | Cryo-EM structures of the *M. tuberculosis* Eσ[B] protomer and dimer (Eσ[B])₂. a** Cryo-EM map and fitted molecular model of Eσ[B] (protomer R1): view from the RNA exit channel (left) and from the main channel (right). The RNAP subunits are colored as indicated on the left. The key structural RNAP modules[68] are indicated. **b** Location of σ[B] in the structure of the R1-R5 dimer (Eσ[B])₂. The RNAP core is shown as a gray semi-transparent molecular surface. The σ[B] subunits assigned to different protomers are colored in deep sky blue and coral. The solved $\sigma_2^B$ domain is shown as a cartoon with cylindrical helices. The unmodeled C-terminal segments of σ[B] (i.e. a1, a2, b1, b2) are shown as molecular surfaces. Green spheres indicate catalytic Mg$^{2+}$ ions. **c** Steric clash between the C-terminal segment of σ[B] and the RNA:DNA hybrid in the active site of RNAP from the R1-R5 dimer. The 5-nt nascent RNA (yellow) and 8-nt template DNA (pink) (from PDB ID 6KON) are shown as cartoons. The b2 density of σ[B] is shown as a molecular surface colored sky blue. **d** Superposition of class I (blue) and class II (coral) R1-R5 dimers.

Supplementary Fig. 9a, b). Two peripheral densities (b1 in R1 and b2 in R5) were buried deeply in the active site cleft of RNAP and occluded the RNA:DNA hybrid path (Fig. 2b, c). Potentially, the a1-a2 density can comprise domain σ[B]4 or domain σ[B]3.1. These two domains contact the β flap and β lobe-2, respectively, in the mature form of the RNAP holoenzyme. However, σ[B]4 interaction with the β flap is incompatible with the dimer model because the σ[B]4 residues 242–260 and 312–323 should clash with the σ[B] region 2 (Supplementary Fig. 9c). Consequently, the atomic model of the σ[B]4 dimer could not be fitted into the cryo-EM density without significant rearrangements of the σ[B]4 structure. Conversely, the atomic model of the σ[B]3.1 dimer perfectly fitted into the density without significant rearrangements (Supplementary Fig. 9b). However, at the current resolution, we could not unambiguously assign the a1–a2 density to σ[B]3.1.

### RNAP conformational flexibility: movements of the β flap

The β flap domain provides an anchoring point for σ4 and paves the RNA exit channel. In the transcriptionally competent RNAP holoenzyme, the β flap tip faced the β′ dock (β′a11, aa 440–495) of the RNA exit channel (open flap) and positioned σ4 relative to σ2 at the optimal distance for promoter −10/−35 element recognition (i.e., 60–64 Å in Mtb Eσ[A]) (Table 1). In Eσ[B], the β flap tip was turned 111° towards the $\sigma_2^B$ and β′ CC that is equal to its 27 Å displacement (closed flap state) (Fig. 3a). The distance between β flap and β′ CC decreased to 46 Å (Table 1). Thus, σ[B]4 could not be correctly positioned for the −35 element promoter binding. Similarly, *Thermus thermophilus* Eσ[A] in complex with the bacteriophage protein gp39 (inhibited state) displays a closed β flap conformation (rotation angle 80°)[32] (Fig. 3a) as well as the terminating elongation complex[33]. In the RNAP octamer, the β flap was captured in the closed conformation by interactions with the σ[B] sub-unit subregion 2.3 and σ[B] C-terminus of the neighboring protomer (Supplementary Fig. 8a). The β flap density was absent in our Mtb RNAP core structure (Supplementary Fig. 2) and in the published cryo-EM structures of *M. smegmatis* RNAP core and holoenzyme[28]. These observations support the notion that the β flap is flexible and can adopt closed/open states if not bound to any partner[34]. Altogether, these results suggest that the β flap oscillation between two utmost conformations is a target for positive (RbpA) and negative (gp39, ρ termination) regulation of transcription.

### RNAP conformational flexibility: movements of the β′ clamp

Published cryo-EM structures of Mtb Eσ[A], free or in complex with ligands: RbpA, CarD, promoter DNA, or fidaxomicin (Fdx), have various clamp states, from open in the Fdx/Eσ[A] complex to closed in RPo[25–27]. Superposition of the Eσ[B] structure with the published models showed that in Eσ[B], the clamp adopted a "fully open" conformation with a rotation angle of 22° relative to σ[A]-RPo (Fig. 3b) and a clamp-β lobe distance of 28 Å (Table 1). This conformation was different from the open and relaxed clamp conformations observed in Eσ[A], RbpA/Eσ[A], and Fdx/Eσ[A] that displayed a clamp rotation angle of 12°–15° (Fig. 3b and Supplementary Fig. 10). In addition, Eσ[B] exhibited different positions of the β′jaw (aa 1037–1116), which moved toward the secondary channel rim-helices (β′ K742-H792) and of the β′ dock, which moved

### Table 1 | Conformational changes in RNAP (distances between Cα atoms in Å)

| Structure | βflap (L820)/ β′CC (L360) | βlobe (T406)/ β′CC (L360) | β1 (G284)/ β′clamp (K123) | PDB ID |
|---|---|---|---|---|
| *Mtb*Eσ[B] | 46 | 49 | 33 | This study |
| *Msmeg* Eσ[A] | UNK | 45 | 25 | 6EYD |
| *Mtb* RbpA/Eσ[A] | 64 | 46 | 26 | 6C05 |
| *Mtb* Eσ[A]- Fdx | 60 | 53 | 29 | 6FBV |
| *Mtb* RbpA/ σ[A]-RPo | 63 | 36 | 16 | 6C04 |

*Mtb* Mycobacterium tuberculosis, *Msmeg* Mycobacterium smegmatis.

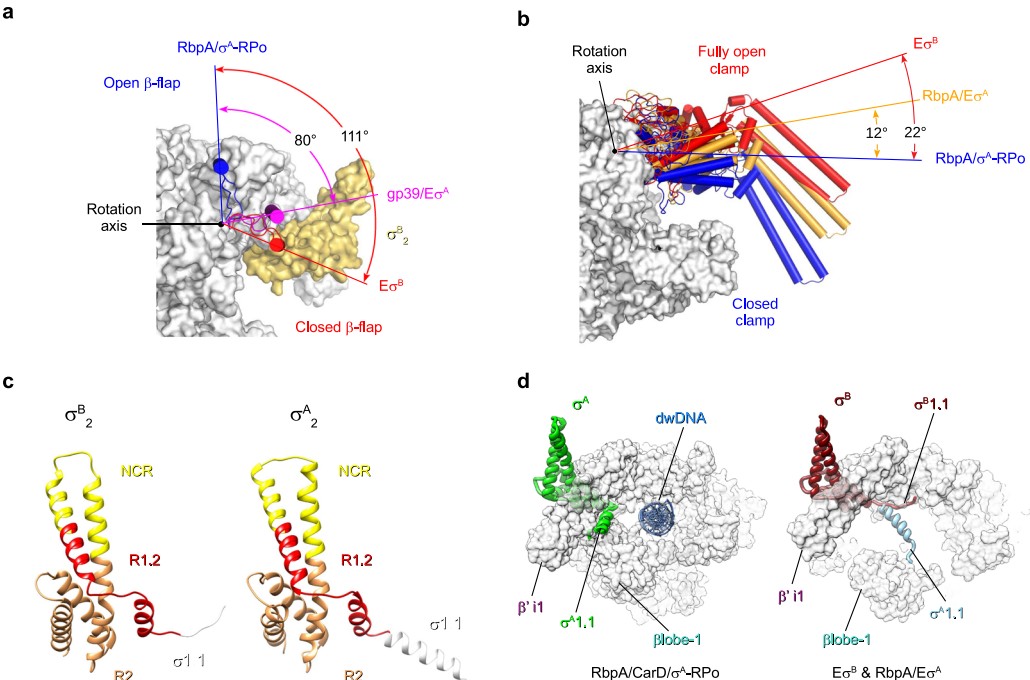

**Fig. 3 | Conformational flexibility of *M. tuberculosis* RNAP.** Motions of the RNAP flap (**a**) and clamp (**b**). RNAP is shown as a molecular surface with the core subunits in gray and σB in yellow. The flap (β flap, β subunit aa 808–832) and clamp (β' subunit aa 1–413, 1219–1245; β subunit aa 1117–1140) domains are shown as cartoons with cylindrical helices. Rotation angles were measured in PyMol as described by[26]. **a** The β flap position in EσB (in red) relative to its position in RbpA/σA-RPo (blue, PDB ID 6C04) and in the *T. thermophilus* gp39/EσA complex (pink, PDB ID 3WOD). **b** The

clamp position in EσB (in red) relative to RbpA/σA-RPo (blue, PDB ID 6C04) and RbpA/EσA (orange, PDB ID 6C05). **c** Comparison of the σB and σA (PDB ID 6C05) structures. The σ subunits are shown as ribbons with domain σ1.1 in white, subregion R1.2 in red, NCR in yellow, R2 in goldenrod. **d** Motions of domain σ1.1. Superposition of σ1.1 in EσB (dark red), RbpA/EσA (light blue) and RbpA/CarD/EσA (green, PDB ID 6EDT). The RNAP core is shown as a molecular surface in gray. σA and σB are shown as ribbons; dwDNA (blue): downstream fragment of promoter DNA duplex.

toward the clamp (Supplementary Fig. 10). The conformational mobility of these domains may potentially affect promoter binding and RNA synthesis[35]. A fully open clamp state was the only clamp conformation observed in our Mtb RNAP sample. We did not detect the partially closed or closed clamp states that were observed by smFRET in *Escherichia coli* Eσ70 [10]. This discrepancy can be due to the lineage-specific properties of Mtb RNAP or to the different buffer composition (e.g., divalent cation concentration) that affect clamp dynamics[12]. We concluded that the fully open-clamp state observed in EσB is characteristic of RNAP inactive state in which the σ subunit regions 3.2 and 4 are unloaded from the RNA exit channel. This immature RNAP conformation may represent an assembly intermediate on the pathway to a transcriptionally active mature conformation of the RNAP holoenzyme.

## Distinct structural signatures of σB and σA

The overall fold of domain $\sigma_2^B$ was similar to that of $\sigma_2^A$ in the published structures of EσA (Fig. 3c). The exception was the N-terminal domain 1.1 that in σB (σB1.1) was stacked to the β' clamp surface in the downstream DNA (dwDNA) channel (Fig. 3d). Conversely, in σA1.1, the α-helix (a.a. 208–223) was located perpendicular to the clamp inside the dwDNA channel and contacted β lobe 1 (Fig. 3d). Thus, σA1.1 hindered the dwDNA access to the main channel of the RNAP holoenzyme. In the σA-RPo structure, σA1.1 was displaced upstream towards β'i1, making the main channel accessible for dwDNA (Fig. 3d). In the EσB structure, σB1.1 did not block the main channel and thus there was no physical barrier for dwDNA entry. This difference may affect RPo formation that is regulated by domain σ1.1[36] and may explain the previously observed different behaviors of EσB and EσA in transcription initiation[23].

Structural sequence alignment of $\sigma_2^B$ and $\sigma_2^A$ (Fig. 4a) showed high sequence similarity in all regions, but for the σ non-conserved region

(NCR). Indeed, in σANCR, four residues are inserted between the α-helices 2 and 3, thus making its tip wider (Fig. 3c). An alignment of 250 actinobacterial σA and σB homologous sequences retrieved in a Blast search showed that σA, but not σB harbored insertions of various lengths in the σNCR tip. In addition, some σB of the genus *Pseudonacardia* presented 5–7 aa insertions in the NCR (Supplementary Fig. 11a). Alignments revealed the σNCR specific patterns (Fig. 4a). Specifically, 89% of σA-NCR harbored conserved tryptophan residue (W283) in α3-helix that was substituted by alanine (A78) in σB. Also, σB-NCR harbored an histidine residue at position 60 (94% of σB sequences) and a leucine residue at position 62 (98% of sequences) that were conserved in σB, but not in σA. Hydrophobic interaction of σB-L62 in the a2-helix with σB-L76 in the a3-helix created a bridge that can stabilize the helix-turn-helix domain of σB NCR.

## The σB subunit domain 2 ties RNAP protomers together

Analysis of the EσB octamer and dimer structures revealed four principal intersubunit interfaces to hold the RNAP protomers together: β flap-σB, β'ZBD-σB, β'i1-σB and σBNCR-σBNCR (Fig. 4b, c, and Supplementary Fig. 11b). All these interfaces included the $\sigma_2^B$ residues responsible for binding to the promoter −10 element and RbpA (Fig. 4a). The invariant W144, from the −10 element recognition W-dyad (subregion 2.3), contacted R830 in the β flap. Thus, the W-dyad was sequestered by interactions with the β flap and stabilized in the "edge-on" conformation that is incompatible with −10 element binding[9]. The residues in the σB subregion 1.2 (E53), σB NCR (Y57, H60, R67) and σB subregion 2.3 (D131, Y132) made contacts with residues R69, R67 and I73 in β'-ZBD. Of these residues, σB Y57 (σA Y258), σB H60 (σA Q261), σB D131 (σA D336), and σB Y132 (σA Y337) were located at the RbpA-binding interface (Fig. 4a, d). Thus we concluded that RbpA-binding and oligomerization are mutually

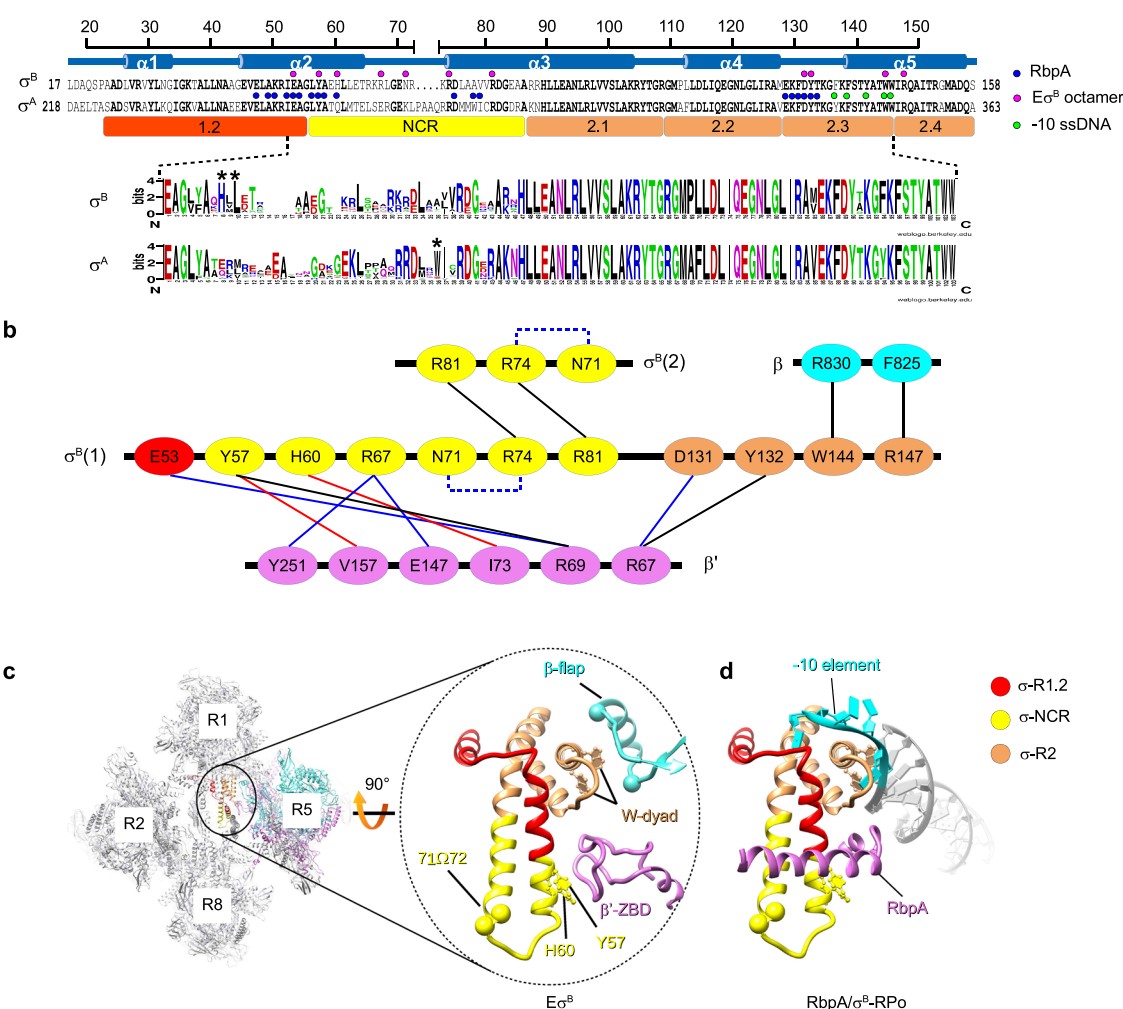

**Fig. 4 | Structure of the Eσᴮ octamer-forming interfaces. a** Schematic representation of the organization of the solved segment of the σᴮ subunit. Top, secondary–structure of σᴮ with the α-helices (α1 to α5) and the structure-based sequence alignment of σᴮ and σᴬ. Dots indicate amino acids that interact with neighboring RNAP protomers in the Eσᴮ octamer structure (magenta) or with RbpA (blue) and promoter -10 element ssDNA (light green) in RbpA/σᴬ-RPo. The evolutionarily conserved subregions 1.2, 2.1, 2.2, 2.3, 2.4, and NCR, are depicted by colored rectangles. Bottom, σᴮ and σᴬ sequence logos generated by Weblogo[69] based on the alignment of the 250 *Actinobacteria* sequences from Uniprot. **b** Schematic representation of the molecular interactions in the Eσᴮ octamer. Residues of the RNAP subunits are presented as ovals and colored according to the color code of **a**. Interactions between residues are shown by lines: π-stacking in black, Van der Waals in red, ionic in blue. Intra-subunit ionic contacts by dashed lines. **c** Interactions holding the RNAP protomers together. Left, cartoon presentation of the Eσᴮ octamer with the RNAP protomer numbers indicated. The zoomed encircled region (labeled Eσᴮ) shows the interactions between σᴮ in the R1 protomer with β′ ZBD and β flap in the R5 protomer. The W-dyad, (i.e. the invariant W144 and W145 residues), interacts with promoter -10 element ssDNA in RPo. Residues Y57 and H60 that contact β′ ZBD are shown as ball and stick models. The Cα atoms of residues 71 and 72 in σᴮ NCR (71Ω72) mark the insertion position in the mutant σᴮ71Ω72. The Cα atoms of the β subunit residues 811 and 825 mark the position of the deletion introduced in the β flap. The σᴮ color codes: subregion 1.2 (aa 27–55) in red, NCR (aa 56–86) in yellow, region 2 (aa 87–158) in sandy brown. **d** Homology model of the RbpA/σᴮ-RPo complex built from the RbpA/σᴬ-RPo model (PDB ID 6C04). Pale green, non-template DNA strand; purple, template DNA strand; magenta, RbpA; gray, DNA; cyan, promoter -10 element.

exclusive events. Residues in σᴮNCR (Y57, R67) also interacted with residues in β′i1 (V157, Y251). Finally, R81 (σᴬ R286) and R74 (σᴬ R279) from two adjacent σᴮ NCR faced each other and could make π-stacking interactions[37] through guanidinium groups stabilized by N71.

## Eσᴮ octamer formation is hindered by RbpA, but not CarD
We used negative stain EM and analytical size-exclusion chromatography (SEC) to explore the conditions for Eσᴮ oligomerization (Fig. 5). Analysis of negatively stained samples showed that Eσᴮ formed octamers spontaneously, within 1 h of mixing the σᴮ subunit with the RNAP core. After 24 h incubation at 4 °C, ~86% of Eσᴮ

molecules were in oligomers (Fig. 5a) migrating as a single peak during SEC (Fig. 5b, marked as (Eσᴮ)₈). Octamer formation occurred starting from RNAP concentrations of 0.1 μM which is several orders of magnitude lower than the bulk RNAP concentration in bacterial cells estimated at 5–28 μM[38,39]. These results, suggest that RNAP octamer assembly may occur in vivo. Next, we explored the capacity to form octamers by the Mtb RNAP core, Mtb Eσᴬ and a chimeric RNAP holoenzyme assembled from Mtb σᴮ and *E. coli* RNAP core (*Eco* Eσᴮ). Analysis of the negatively stained images and SEC profiles showed that none of these proteins could form octamers in our experimental conditions (Fig. 5b–d). Although, Mtb RNAP displayed a high propensity for dimer formation (Fig. 5b). Thus, we concluded

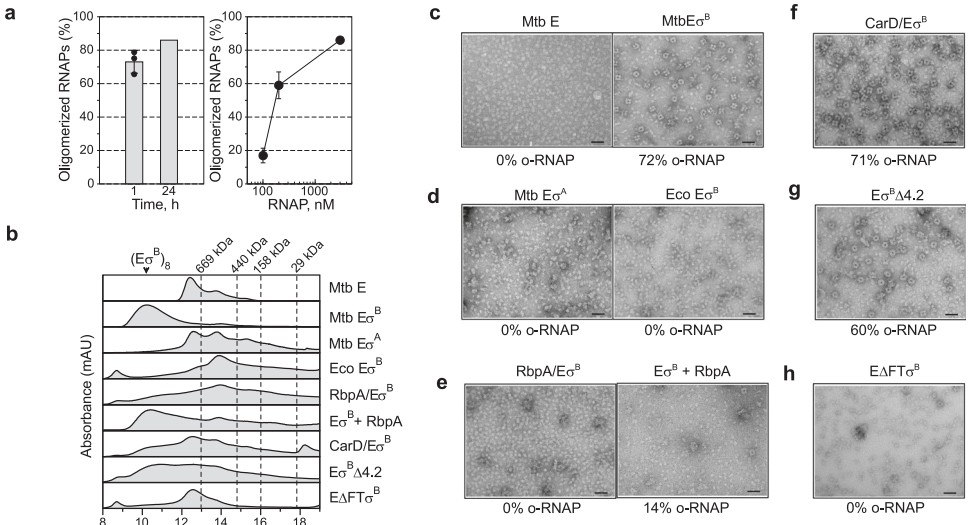

**Fig. 5 | Regulation of the Eσ^B octamer assembly in vitro. a** Time (left panel) and concentration (right panel) dependence of the Eσ^B octamer assembly. Bar graphs show the percentage of RNAP molecules assembled in octamers relative to the total number of RNAP molecules visualized on the EM grid. Values were calculated from the negatively stained images by counting particles using the EMAN2 e2boxer module. When indicated, data are presented as mean values ± SD calculated from $n = 3$ (bar graph) and $n = 4$ (line graph) representative images. **b** Assessment of the RNAP oligomerization state by SEC. Elution profiles of the RNAP core (E) in complex with a different set of transcription factors (indicated on the right) match combinations shown in **c**–**h**. The octamer peak is indicated as (Eσ^B)_8. Dashed lines indicate the position of the molecular weight markers: 669 kDa, thyroglobulin; 440 kDa,

ferritin; 158 kDa, aldolase; 29 kDa, carbonic anhydrase. **c** Negatively stained images of the Mtb RNAP core (Mtb E) and Mtb RNAP holoenzymes (Mtb Eσ^B). **d** Negatively stained images of Mtb Eσ^A and of the hybrid RNAP holoenzyme assembled from σ^B and *E. coli* RNAP core (Eco Eσ^B). **e** Negatively stained images of Mtb Eσ^B. RbpA was added to E before assembly with σ^B (RbpA/Eσ^B) and after the octamer formation (Eσ^B + RbpA). **f** Negatively stained images of Mtb Eσ^B formed in presence of CarD (CarD/Eσ^B) added to E before assembly with σ^B. Negatively stained images of mutant Eσ^B harboring a deletion in the σ^B subregion 4.2 (Eσ^BΔ4.2) (**g**) or a deletion in the β flap (EΔFTσ^B) (**h**). Scale bar in **c**–**h** = 50 nm. Experiments in **c**–**h** were repeated independently at least twice with similar results. Source data are provided as a Source Data file.

---

that the specific structural features of σ^B and of Mtb RNAP core are essential for spontaneous Eσ^B oligomerization. The specific structural features of σ^A, such as the insertion in the σ^ANCR and the bulky side chains of the σ^A-specific residues (e.g., W283), should interfere with the σ^ANCR-σ^ANCR interactions and hinder oligomerization. Next, we asked whether RbpA and CarD affect Eσ^B oligomerization (Fig. 5b, e, f). As RbpA binds to σNCR, it should interfere with the RNAP-octamer formation (Fig. 4e). Indeed, addition of a 2-fold molar excess of RbpA to the RNAP core before the addition of σ^B, hindered octamer formation while addition of RbpA to pre-formed Eσ^B octamer induced its dissociation (Fig. 5b, e). As CarD interacts with RNAP β lobe 1, it should not interfere with the octamer-forming interactions of RNAP subunits. Indeed, the negatively stained images show that addition of CarD to the RNAP core before the addition of σ^B did not prevent octamer formation (Fig. 5b, f). Yet, SEC showed a decrease in the amount of octamers in the presence of CarD, which indicates that CarD affects oligomerization or stability of octamer. These results suggest that RbpA can selectively regulate σ^B activity by modulating RNAP oligomerization.

### Interaction of the β flap with σ^B stabilizes the Eσ^B octamer
In the Eσ^B octamer structure, the σ^B subregions 2.3 and 2.4 interact with the β flap. To explore whether the β flap and its binding partner σ4 affected octamer assembly, we constructed RNAP mutants in which residues 811-825 in the β flap tip (Mtb E^ΔFTσ^B) and residues 252-323 in the σ^B domain 4.2 (Mtb Eσ^BΔ4.2) were deleted. Analysis of the SEC profiles (Fig. 5b) and the negatively stained images (Fig. 5g) showed that RNAP harboring σ^BΔ4.2 formed octamers, suggesting that σ^B4.2 is not essential for oligomerization. Conversely, deletion of the β flap tip (Mtb E^ΔFTσ^B) inhibited octamer formation (Fig. 5b, h), suggesting that its interaction with the σ^B region 2 is critical for oligomerization. Altogether, these results suggest that σ^B4.2 does not make contacts with the β flap in the octamer.

### Role of σ^B-Y57 and σ^B-H60 in oligomerization and transcription
The σ^BNCR residues Y57 and H60, which make contacts with β'ZBD in the octamer, also are implicated in RbpA-binding[40]. To explore their functional role, we constructed two σ^B mutants harboring the Y57A and H60A substitutions. We used negative stain EM to visualize the RNAP holoenzymes assembled with σ^B-Y57A and σ^B-H60A. Analysis of the images demonstrated that both substitutions abolished octamer formation (Fig. 6a), although unstructured oligomers/aggregates were present. SEC also showed no octamer formation by the mutant RNAPs (Fig. 6b).

Run-off transcription assays performed with the mutant Eσ^BY57A and Eσ^BH60A holoenzymes demonstrated that Y57, but not H60, was required for transcription initiation at the RbpA-dependent *sigA*P promoter and at the RbpA-independent synthetic *sigA*Pext-10 promoter (Fig. 6c). The Y57A substitution abolished stimulation of transcription by RbpA at the *sigA*P promoter, in agreement with published data on σ^A[40]. Unexpectedly, Y57A did not abolish stimulation of transcription by RbpA at the *sigA*Pext-10 promoter, suggesting that Y57 is not essential for RbpA-binding to RNAP. We concluded that extensive contacts between RbpA and the σ subunit can compensate for the lack of the σ^BY57A-RbpA contact (Fig. 4a). Furthermore, Y57A abolished run-off RNA synthesis from the *sigA*Pext-10 promoter in the absence of RbpA, suggesting that this residue is implicated in transcription initiation also through RbpA-unrelated mechanisms. On the basis of the RbpA/σ^A-RPo structure, Y57 is positioned too far from promoter DNA (~20 Å) to affect RNAP-DNA binding directly. It may stabilize an overall fold of σ domain 2, and thus stimulates its interaction with the promoter −10 element.

### The σ^B NCR tip affects oligomerization and RPo formation
We showed that the σ^A subunit cannot induce octamer formation (Fig. 5b, d). As that σ^BNCR is implicated in three of four intersubunit interfaces holding RNAPs in the octamer, we hypothesized that the

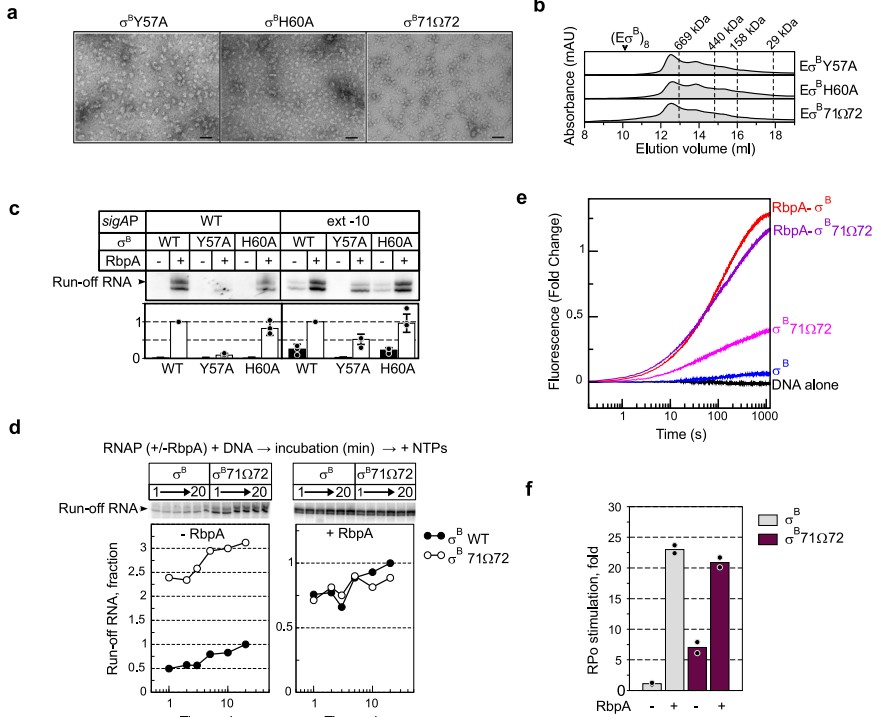

**Fig. 6 | Role of σB NCR in octamer assembly and RPo formation. a** Negatively stained images of EσB harboring the following mutant σ subunits: σB-Y57A, σB-H60A, and σB71Ω72. Scale bar = 50 nm. Experiments were repeated independently twice with similar results. **b** Assessment of the mutant RNAPs oligomerization state by SEC. Elution profiles of the RNAP core (E) in complex with σB-Y57A, σB-H60A, and σB-71Ω72. Expected position of the octamer peak is indicated as (EσB)8.
**c** Transcriptional activity of Mtb RNAP harboring σB (WT) and σB mutants (Y57A, H60A). Transcription was initiated at the wild-type sigAP (WT) and sigAPext-10 promoters either without (-) or with (+) RbpA. Gels show [α32P]-labeled run-off RNA products. Bar graph shows the band quantification. For each panel, the run-off RNA amounts were normalized to the RNA amount synthesized in the presence of RbpA.

Data are presented as mean values ± SD of three independent experiments.
**d** Kinetics of promoter binding by the EσB71Ω72 mutant in run-off transcription assays using the sigAPext -10 promoter variant. The experimental scheme is shown at the top. Inserts show run-off RNA products. Graphs show the normalized amounts of run-off RNA (σB71Ω72 vs σB) as a function of time. **e** Fluorescence fold-change during RPo formation kinetics by EσB and EσB71Ω72 on the Cy3-labeled sigAPext-10 promoter without or with RbpA. **f** RPo fractions at equilibrium in the time-resolved fluorescence assay shown in **d**. Values were normalized to the value for EσB. Data points from two technical replicates are shown. Mean values are presented as bar graphs. Source data are provided as a Source Data file.

insertion in the σANCR tip (Figs. 3c and 4a) compromises intersubunit interactions and may hinder RNAP oligomerization. To test this hypothesis, we introduced a σA fragment (residues PAAQ) between residues 71 and 72 of σB (σB 71Ω72)(Fig. 4a, c). Analysis of the RNAP holoenzyme harboring the mutant σB 71Ω72 by negative stain EM and SEC showed that the insertion abolished octamer formation (Fig. 6a, b).

Oligomerization captures RNAP in the inactive state and is expected to inhibit transcription, while its suppression should stimulate transcription. To assess the oligomerization effect on transcription initiation, we used σB 71Ω72 and two complementary methods: single-round run-off transcription assay[24] and fluorescent assay to follow RPo formation kinetics in real-time[41,42]. To minimize the RNAP dependence on RbpA, we used as template the sigAPext-10 promoter that forms RPo by EσB without RbpA[24]. We monitored run-off RNA synthesis at different time points after mixing RNAP with promoter DNA. (Fig. 6d). Without RbpA, the EσB 71Ω72 mutant (monomeric state) displayed ~3-fold higher transcriptional activity than EσB (octameric state). In the presence of RbpA (monomeric state), EσB and EσB 71Ω72 showed similar transcriptional activity. Therefore, we concluded that insertion in σB NCR stimulates transcription initiation possibly by inhibiting octamer formation. As the σB NCR residues 71 and 72 are located too far from the promoter, they cannot affect initiation directly through interaction with promoter DNA, as shown for *E. coli* σ70 NCR[43].

To assess whether the 71Ω72 insertion affected directly RPo formation kinetics, we used the sigAPext-10 promoter with the Cy3 dye

tethered to the guanine at position +2 of the non-template DNA strand[24]. Next, we followed Cy3 fluorescence increase upon RNAP binding to the promoter, which is a characteristic of RPo formation (Fig. 6e, f). In agreement with the previous findings[42], the reaction kinetics was best fitted by a triple-exponential equation with three phases: fast, medium, and slow (Table 2). The fast phase was over in the first 15 -20 s and may reflect perturbations in the system during the initial promoter melting step. The next two steps were slow ($t_{1/2}$ between 0.6 and 2.5 min) and reflected isomerization of the closed promoter complex (RPc) to RPo[42]. All three kinetics of RPo formation by RNAPs in the monomeric state (RbpA/EσB, EσB71Ω72 and RbpA/EσB 71Ω72) displayed similar fractional amplitudes and rate constants for the intermediate and slow phases. The kinetic constants of RPo formation by RNAP in the octameric state (EσB) were different (Table 2). Specifically, the fractional amplitudes and rate constants for the intermediate and slow phases could not be distinguished probably due to the low signal amplitude. Furthermore, RbpA accelerated by ~3-fold RPo formation by EσB, but not by EσB 71Ω72. Without RbpA, the mutant EσB 71Ω72 formed more RPo (~6-fold) than EσB (Fig. 6f). In the presence of RbpA, the amount of RPo was the same for wild type and mutant RNAPs, in agreement with the results of the transcription assay (Fig. 6d). Thus, we concluded that the insertion in σBNCR stimulates RPo formation by increasing the effective concentration of RNAP monomer available for promoter binding. Mutant EσB 71Ω72 is still susceptible to RbpA activation, in agreement with the dual-mode activation mechanism in which RbpA promotes σ loading to RNAP[6] and stabilizes RNAP interaction with promoter DNA[40].

**Table 2 | Kinetic constants of RPo formation on the *sigA*Pext-10(+2Cy3) promoter**

| Condition | Fast $k_1$ (s$^{-1}$) | Intermediate $k_2$ (s$^{-1}$) | Slow $k_3$ (s$^{-1}$) | Fast $A_1$,fraction | Intermediate $A_2$, fraction | Slow $A_3$, fraction | Total $\Sigma A_i$, FC |
|---|---|---|---|---|---|---|---|
| E$\sigma^B$ | 0.049 | 0.005 | 0.005 | 0.253 | 0.373 | 0.373 | 0.064 |
| RbpA/E$\sigma^B$ | 0.078 | 0.015 | 0.003 | 0.123 | 0.372 | 0.505 | 1.291 |
| E$\sigma^B$ 71Ω72 | 0.083 | 0.018 | 0.002 | 0.212 | 0.348 | 0.440 | 0.405 |
| RbpA/E$\sigma^B$ 71Ω72 | 0.052 | 0.011 | 0.002 | 0.262 | 0.303 | 0.436 | 1.177 |

$A_1$, $A_2$, $A_3$ are fractional amplitudes for each phase and $\Sigma A_i$ is the total amplitude of the process at equilibrium.

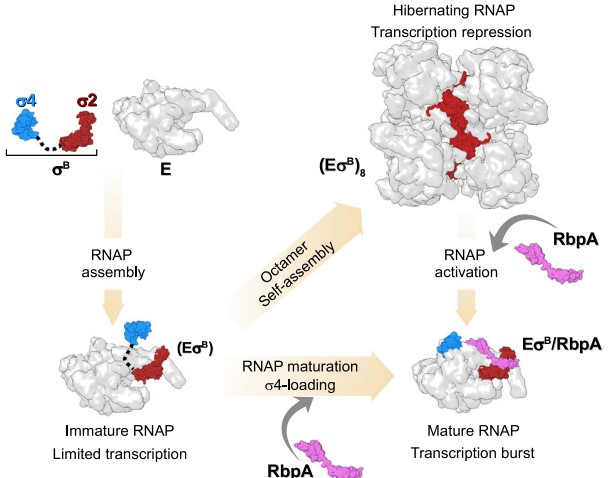

**Fig. 7 | Model depicting transcription regulation through Eσ$^B$ holoenzyme assembly and oligomerization.** A pathway to Mtb Eσ$^B$ maturation. The pathway starts with the formation of the immature RNAP holoenzyme from free σ$^B$ [σ2 domain (dark red) and σ4 domain (cyan)] and RNAP core (E, depicted as a gray molecular surface). In the immature Eσ$^B$, the σ$^B$ domain 4 remains unloaded from the RNA exit channel, thus making it prone to oligomerization. Binding of RbpA (depicted as a pink molecular surface) to Eσ$^B$ prevents oligomerization and converts Eσ$^B$ to the transcriptionally active, mature form with the σ$^B$ domain 4 loaded in the RNA exit channel. Binding of RbpA to the (Eσ$^B$)$_8$ octamer induces its dissociation.

## Discussion

In this study we show that Mtb RNAP harboring the group II σ$^B$ factor can spontaneously oligomerize into symmetric octamers in which RNAP is captured in an inactive conformation with the domain σ4 unloaded from the RNAP core. RNAP oligomerization is reversed by the global transcriptional activator RbpA that, according to smFRET and biochemical studies[6,44] induces σ4 domain loading onto RNAP (see model in Fig. 7). In bacterial cells, σ factors compete for a general RNAP core pool[38]. Our findings suggest that σ$^B$ inhibits its own activity by sequestering the RNAP core and may act as a repressor of σ$^A$-dependent genes by decreasing the effective concentration of free RNAP. In support to this hypothesis, both σ$^B$ and RbpA are implicated in the regulation of transcription in the *M. tuberculosis* starvation and dormancy models[45,46]. Thus, contrary to *sigA*, the *sigB* gene expression was shown to be upregulated 6-fold concurrently with RbpA (a 9-fold induction)[45–47]. We speculate that the Eσ$^B$ octamer may serve as a storage of hibernating Eσ$^B$ that can be rapidly converted to the active form by RbpA during stress. Eσ$^B$ octamer disassembly leads to a burst in the effective RNAP concentration and consequently should boost transcription yields (Fig. 7). From our results, σ$^B$NCR emerges as a major determinant of Mtb Eσ$^B$ octamer assembly. Furthermore, our results suggest that small variations in the σNCR structure can have inhibitory and also stimulatory effects on RPo formation.

To our knowledge, no structure of octamers formed by RNAPs has been described up to date. Early biochemical studies reported formation of octamers by the *E. coli* RNAP core in the presence of Mg$^{2+}$

ions and dimers by the *E. coli* Eσ$^{70}$ holoenzyme[48,49]. The structure of these complexes has not been characterized. Furthermore, neither *E. coli* nor Mtb RNAP core formed octamers in our experimental conditions. Self-assembly of proteins into high-order structures is essential for regulating all processes in cellular organisms[50]. For instance, it has been shown that the formation of symmetric tetramers by influenza virus RNA-dependent RNA polymerase (RdRP) is important for the viral life cycle regulation[51]. Yet, only two cases of regulation that implicate self-assembly of the cellular gene expression machinery into supramolecular structures have been thoroughly described: dimerization of ribosomes in bacteria[52,53] and dimerization of RNAP I in yeast[54–57]. In both cases, dimerization was induced by nutrient deprivation and employed as a regulatory mechanism to control ribosome biogenesis and protein synthesis during cell hibernation. Recently, it has been proposed that dimerization of the *Bacillus subtilis* RNAP core in complex with the HelD helicase is a hibernation mechanism[58]. There is an amazing convergence in the mechanisms of RNAP inactivation between the evolutionary distant yeast RNAP I and Mtb Eσ$^B$. Like in Mtb Eσ$^B$, in inactive RNAP I dimers, the clamp adopts an open state[56]. Furthermore, like RbpA, the eukaryotic initiation factor Rrn3, interacts with the RNAP I regions involved in dimer formation and converts RNAP I dimers to initiation-competent monomers[55,59] Thus, RNAP oligomerization controlled by transcription factors emerges as a general regulation mechanism for gene repression in the different kingdoms of life. An exciting direction for future studies will be to explore RNAP octamer formation in live cells and to determine its function.

## Methods
### Proteins and DNA templates
The 6xHis-tagged Mtb RNAP core was expressed in BL21 DE3 *E. coli* cells transformed with the pMR4 plasmid and purified by Ni$^{2+}$-affinity chromatography[17] followed by purification by anion-exchange chromatography on MonoQ 5/50GL (GE Healthcare). In brief, 15 g of bacteria were resuspended in 200 ml of lysis buffer (40 mM Tris-HCl pH 7.9, 500 mM NaCl, 5% Glycerol, 10 μM ZnCl$_2$, 0,2 mM β-mercaptoethanol, 4 tablets of cOmplete™, EDTA-free Protease Inhibitor Cocktail (Roche), 0.1 mg/ml lysozym). Cells were disrupted by sonication and the lysate, cleared by centrifugation at 9,000 g (40 min, 4 °C), was loaded on 5 × 5 ml HisTrap FF columns (GE Heathcare) equilibrated with 40 mM Tris-HCl pH 7.9, 500 mM NaCl, 5% glycerol, 0.2 mM β-mercaptoethanol. Mtb RNAP core was eluted with 300 mM imidazole. Mtb RNAP sample was dialyzed at 4 °C overnight against buffer A (20 mM Tris-HCl pH 7.9, 500 mM NaCl, 5% glycerol, 0,2 mM EDTA, 0.01% Tween20, 10 mM ZnCl$_2$, 0,2 mM β-mercaptoethanol). Next, RNAP sample was dialyzed at 4 °C for 2 h against buffer B (20 mM Tris-HCl pH 7.9, 50 mM NaCl, 5% Glycerol, 0.2 mM EDTA, 0.2 mM β-mercaptoethanol, 0.01% Tween20) and loaded on MonoQ 5/50GL column (GE Heathcare) equilibrated with buffer B. Mtb RNAP core was eluted in linear gradient of NaCl (200–480 mM). Collected peak fractions were pooled, concentrated by Amicon® Ultra-15 Centrifugal Filter Unit (Millipore), supplemented with 20% glycerol, and stored at −80 °C. The 6xHis-tagged *E. coli* RNAP core was expressed in BL21 DE3 *E. coli* cells transformed with the pVS10 plasmid and purified as described above for Mtb RNAP. The gene encoding for Mtb CarD (Rv3583c) was PCR-amplified from H37Rv genomic DNA (NR-14867,

BEI Resources) and cloned between the *Nde*I and *Hind*III sites into the pET28a vector under the N-terminal 6xHis-tag. The 6xHis-tagged Mtb CarD, RbpA, and the $\sigma^A$ subunit were expressed and purified by chromatography on HisTrap HP (GE Healthcare) Ni$^{2+}$-affinity columns as described. The 6xHis-tagged Mtb $\sigma^B$ subunit, $\sigma^B$ in which residues 252–323 were deleted ($\sigma^B\Delta4.2$), and $\sigma^B$ with the insertion in NCR ($\sigma^B71\Omega72$) were purified by chromatography on HiTrap TALON (GE Healthcare) columns. To construct the mutant $\sigma^B$ 71Ω72, residues N71 and R72 of the $\sigma^B$ subunit were replaced by six $\sigma^A$ residues (segment 272-KLPAAQ-277) using the Quick Change Lightening site-directed mutagenesis kit (Agilent). Residues 811-825 in the Mtb RNAP β flap were deleted using Quick Change II XL site-directed mutagenesis kit (Agilent). Variants of the wild-type *sigA*P and *sigA*Pext-10 (harboring the $T_{-17}G_{-16}T_{-15}G_{-14}$ motif) promoters 5′-end-labeled with fluorescein and the *sigA*Pext-10 promoter internally labeled with Cy3 at position +2 of the non-template DNA strand were prepared as described[24]. Sequences of primers used for CarD cloning and for *sigA*P promoter fragments preparation are provided in Supplementary Table 2

## Cryo-EM sample preparation

To assemble the Eσ$^B$ holoenzyme, 3.4 µM Mtb RNAP core and 3.74 µM $\sigma^B$ in transcription buffer (40 mM HEPES pH 8.0, 50 mM NaCl, 5 mM MgCl$_2$, 5% glycerol) were incubated at 37 °C for 5 min. Next, samples were dialyzed in 10 µl drops on 0.025 µm MF-Millipore membrane filters (VSWP) against dialysis buffer (20 mM HEPES pH 8.0, 150 mM NaCl, 5 mM MgSO$_4$) at 22 °C for 45 min. About 3 µl of sample (3 µM Eσ$^B$ final concentration) was spotted on a Quantifoil R2/2 200 Mesh holey carbon grids which were glow-discharged for 10 s using the PELCO easiGlow system (Ted Pella). Grids were flash-frozen in liquid ethane using Vitrobot Mark IV (FEI) at 20 °C and 95–100% of humidity.

## Cryo-EM data acquisition

Data were collected using a spherical aberration (Cs) - corrected Titan Krios S-FEG instrument (FEI) operating at 300 kV acceleration voltage and equipped with a Gatan K2 Summit direct electron detector (Gatan, Warrendale, PA). Automatic image acquisition was carried out using the EPU software (FEI) in a super-resolution mode at a nominal magnification of ×105,000 with a pixel size of 0.55 Å. Movies (31 frames) were collected at an exposure rate of 6.2 e$^-$/Å$^{-2}$/s and a total electron dose of 49.6 e$^-$/Å$^{-2}$ over a nominal defocus range from −0.5 to −5.0 µm.

## Cryo-EM data processing

Movie frames were aligned, dose-weighted, binned by two, and averaged using Motioncor2[60]. Movie sums were used for contrast transfer function (CTF) estimation with Gctf[61]. A 3064 dose-weighted movie sums were used in the subsequent image-processing steps. About 100 particles comprising monomers and oligomers were manually picked in cryoSPARC[62] and subjected to 2D classification to create templates for automatic picking.

**Mtb RNAP core reconstruction using cryoSPARC.** A total set of 721,752 particles that included RNAP monomers underwent several 2D classification rounds. A cleaned dataset of 153,953 particles was used in the ab-initio reconstruction to compute the initial model. The ab-initio model was used as reference for the 3D heterogeneous refinement and classification. Two 3D classification rounds produced a cleaned set of 55,008 particles that were used for the homogeneous refinement at 4.19 Å. Then, the local non-uniform refinement resulted in an improved cryo-EM map refined at 4.08 Å.

**Mtb Eσ$^B$ octamer and protomer reconstructions using cryoSPARC.** A dataset of 254,380 particles that included RNAP oligomers underwent several 2D classification rounds to produce a clean dataset of 115,112 particles. The best class averages that included 82,893 particles were used in the reference-free ab-initio heterogeneous

reconstruction to produce the initial model. The homogeneous non-uniform refinement of the initial map using 115,112 particles without applying symmetry resulted in a $C_1$-map at a nominal resolution of 6.13 Å. The dataset of 115,457 particles was used for the non-uniform refinement with imposed $D_4$ symmetry to compute the $D_4$-map at a nominal resolution of 4.39 Å. The dataset of 115,112 subtracted particles and the masked $C_1$-map were used for the local non-uniform refinement to calculate the maps of the RNAP monomer and RNAP dimer at a nominal resolution of 3.84 Å and 4.36 Å, respectively The dataset of 115,112 subtracted particles underwent 3D heterogeneous refinement using the R1-R5 RNAP dimer map as reference. The 3D classification produced two RNAP dimer classes with different conformations: class 1, which included 66,519 particles, resolved at 4.38 Å, and class 2, which included 48,593 particles, resolved at 6.75 Å.

**Mtb Eσ$^B$ protomer reconstruction using RELION.** A dataset of 94,191 particles was used in RELION (version 1.4) 3D refinement to compute the $D_4$-map of the Eσ$^B$ octamer at resolution of 6.3 Å. Then, each particle 2D image was replicated and 3D-rotated according to the $D_4$ point group symmetry and the re-projected density of all but one Eσ$^B$ protomer was subtracted. The resulting expanded dataset of 753,528 subtracted particles was used in alternate cycles of asymmetric 3D classification and 3D local refinement focused on the region occupied by a single Eσ$^B$ protomer. Refinement was first focused on the RNAP core density devoid of the β′ clamp. A $D_4$-map of the octamer, from which the focused region was masked out, was used for the signal subtraction. The focused region of the RNAP core, which represented the least resolved peripheral zones in the $D_4$-map of the Eσ$^B$ octamer (Supplementary Fig. 4a), was reconstructed at 3.6 Å resolution (Supplementary Fig. 7). Next, the β′ clamp and $\sigma^B$ subunit were included in the focused refinement. The signal of seven of the eight β′ clamp/$\sigma^B$ re-projected densities was subtracted from the particles projections based on the initial $D_4$-map of the Eσ$^B$ octamer and using an appropriate mask. For each particle projection, the signal of the remaining RNAP core subunits outside the focused region was subtracted using seven successive re-projections based on the 3.6 Å RNAP core reconstruction and the angular parameters obtained in the preceding 3D refinement. Two out of six 3D classes were selected and locally refined at a resolution up to 3.9 Å and 4.19 Å, respectively (Supplementary Fig. 7).

## Model building and refinement

The coordinates of the Mtb Eσ$^A$ holoenzyme (PDB ID 6FBV) were used as starting model. The $\sigma^A$ subunit was replaced by $\sigma^B$ using homology modeling in Modeller[63]. The model was fitted to cryo-EM maps in UCSF Chimera[64] and was manually modified in Coot[65]. Several cycles of real-space refinement, using secondary-structure restrains and geometry optimization, were performed in Phenix[66] using the R1-R5 RNAP dimer and R1 RNAP protomer maps. To improve the $\sigma^B$ subunit model, it was real-space refined separately in Phenix with the RNAP octamer $D_4$-map. The final Eσ$^B$ model was assembled and modified in Coot. The R1-R5 RNAP dimer model was assembled from two copies of Eσ$^B$ rigid body refined in Phenix. The Eσ$^B$ octamer model was built by applying NCS operators to the Eσ$^B$ protomer model in Phenix followed by modification in Coot.

## Negative stain EM sample preparation and data acquisition

100–400 nM RNAP core was mixed with threefold molar excess of the σ subunit in the transcription buffer described above. When indicated, RbpA and CarD (3-fold molar excess) were added to the RNAP core. Samples were incubated at 22 °C for 10 min. The reaction mixtures were dialyzed for 1 h, as described for the cryo-EM samples. Then, 3 µl of mixture was spotted on a Formvar/Carbon copper 200 mesh grids (Electron Microscopy Sciences) glow-discharged for 10 s. Grids were stained with uranyl acetate (1% w/v). Images were collected using an

120 kV JEOL 1200 EX II EM equipped with an EMSIS Quemesa 11Mpixels camera with a nominal magnification of ×50,000 and pixel size 2.86 Å. Particles were counted using the EMAN2 e2boxer module[67].

## Analytical size-exclusion chromatography

To assess oligomeric states of RNAP, about 200 pmole of E and Eσ holoenzymes formed with twofold molar excess of σ$^A$, σ$^B$ variants, RbpA and CarD in 120 μl of transcription buffer were incubated at 24 °C for 1 h. Mtb E was first mixed with RbpA and CarD and then supplemented with the σ$^B$ subunit. To test the effect of RbpA on the octamer dissociation, Mtb E was incubated with σ$^B$ overnight at +4 °C, then supplemented with RbpA and incubated at 24 °C for 1 h. Next, a 100 μl of the sample was applied to Superose® 6 Increase 10/300 GL column (Cytiva) equilibrated with 10 mM Tris-HCl pH 8.0, 150 mM NaCl, 5 mM MgCl$_2$, 5% glycerol and run at 0.5 ml/min flow rate.

## In vitro transcription assay

In multiple-round transcription assays, 100 nM RNAP core was mixed with 300 nM σ$^B$ and 300 nM RbpA in 10μl of transcription buffer and incubated at 37 °C for 5 min. 50 nM of promoter DNA was added and incubated at 37 °C for 10 min. Transcription was initiated by adding 50 μM/each of ATP, GTP, CTP, 5 μM of UTP, and 0.5 μM of [$^{32}$P]-UTP, and performed at 37 °C for 5 min. In single-round kinetics run-off assays, 200 nM RNAP core was mixed with 600 nM σ$^B$ with or without 600 nM RbpA in 10μl of transcription buffer and incubated at 37 °C for 5 min. Samples prepared without RbpA were incubated at +4 °C overnight to reach the maximum yield of RNAP octamer. After the addition of 100 nM of *sigA*Pext-10 promoter, DNA samples were incubated at 37 °C for 1, 2, 3, 5, 10, or 20 min. Transcription was initiated by adding 100 μM/each of CTP, GTP, ATP, 10 μM of UTP, 0.5 μM of [$^{32}$P]-UTP, and 0.1 mg/ml poly(dI·dC). Reactions were incubated at 37 °C for 3 min. RNA transcripts were analyzed by denaturing 18% PAGE/7 M urea gels. Gels were scanned with a Molecular Dynamics STORM Imager and quantified by ImageQuant software.

## Real-time fluorescent assay of RPo formation

Data were acquired using a SF-61 DX2 stopped flow spectrophotometer (TgK Scientific UK) with a shot volume of 100 μl, excitation at 535 nm and emission at 570 nm. 200 nM RNAP core was mixed with 1 μM of σ subunit with or without 1 μM RbpA in 100 μl of transcription buffer and incubated at 37 °C for 5 min. Samples prepared without RbpA were incubated at +4 °C overnight. Protein samples were diluted 4-fold in transcription buffer containing 0.1 mg/ml BSA immediately before mixing with promoter DNA. Experiments were initiated by mixing equal volumes of 50 nM RNAP and 10 nM Cy3-labeled *sigA*Pext-10 promoter in a transcription buffer containing 0.1 mg/ml BSA. The final RNAP concentration was 25 nM and the DNA concentration was 5 nM. Data were collected at 30 °C for 20 min. Two consecutive shots were performed and averaged. Each dataset was normalized to the fluorescence signal value at equilibrium and plotted as the fluorescence fold change (FC), where $FC = (F - F_o)/F_o$. $F_o$ is the signal for DNA alone and $F$ is the signal for RNAP-bound DNA. Values from two experiments were averaged and fitted using the Grace software (v. 5.1.25) with the triple-exponential equation $FC_t = A_0 + A_1 \cdot exp(-k_1 \cdot t) + A_2 \cdot exp(-k_2 \cdot t) + A_3 \cdot exp(-k_3 \cdot t)$ where $FC_t$ is the total fluorescence change.

## Reporting summary

Further information on research design is available in the Nature Portfolio Reporting Summary linked to this article.

## Data availability

The data that support this study are available from the corresponding authors upon reasonable request. Cryo-EM density maps reported in this paper have been deposited in the Electron Microscopy Data Bank

(EMDB) with accession codes EMD-13579 (Mtb Eσ$^B$ protomer), EMD-14696 (Mtb Eσ$^B$ protomer), EMD-13817 (Mtb Eσ$^B$ octamer $D_4$-map), EMD-14697 (Mtb Eσ$^B$ octamer $D_4$-map), EMD-13829 (Mtb Eσ$^B$ dimer), EMD-14378 (Mtb Eσ$^B$ dimer class 1), EMD-14974 (Mtb Eσ$^B$ dimer class 2), EMD-14560 (Mtb RNAP core). Model coordinates have been deposited in the Protein Data Bank (PDB) with accession codes 7PP4 (Mtb Eσ$^B$ protomer), 7ZF2 (Mtb Eσ$^B$ protomer), 7Q4U (Mtb Eσ$^B$ octamer), 7Q59 (Mtb Eσ$^B$ dimer), 7Z8Q (Mtb RNAP core). The publicly available datasets with PDB accession codes 6FBV, 6EYD, 6C04, 6C05, 6KON, 3WOD, 6EDT were used in this study for figure preparation and data analysis. Source data are provided with this paper.

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

## Acknowledgements

We thank Irina Artsimovitch and Mickael Blaise for critical reading of the manuscript and discussion. We thank Jean-Paul Leonetti for help with figure preparation and discussion. We thank Franck Godiard for the negative stained EM data collection at the MEA platform, University of Montpellier, Montpellier, France. We thank Alexander Myasnikov, Jean-Francois Menetret, Julio Ortiz for assistance with grid preparation and cryo-EM data collection at the Center for Integrative Biology, IGBMC, Strasbourg, France. Funding from the French National Research Agency [MycoMaster, ANR-16-CE11-0025] to K.B. and P.B. Support from Instruct-ERIC (PID: 1309) to K.B. Support from CNRS (Frozen, Mission pour l'Interdisciplinarité, AAP Interne 2017) to C.L. The CBS is a member of the French Infrastructure for Integrated Structural Biology (FRISBI), a national infrastructure supported by the French National Research Agency (ANR-10-INBS-05).

## Author contributions

Conceptualization, K.B.; methodology, K.B., P.B., and S.T.; software, S.T.; investigation, Z.M., R.K.V., C.L., J.L.K.H., P.B., and K.B; formal Analysis, C.L., Z.M., L.C., S.T., and K.B.; validation, K.B., and P.B. writing, K.B.; supervision, K.B., and P.B.; funding acquisition, K.B., P.B., and C.L.

## Competing interests

The authors declare no competing interests.
