## [Peer Review File · Nature Communications]

Structural basis of the mycobacterial stress-response RNA polymerase auto-inhibition via oligomerizationReviewers' Comments:

Reviewer #1:

Remarks to the Author:

Oligomerization of virus RdRP, *Bacillus subtilis* RNAP, eukaryotic Pol I and human mitochondrial RNAP previously have been reported as a means to regulate their transcription activity. In this study, the authors reported an auto-inhibition mechanism of *Mycobacteria tuberculosis* RNAP-sigB holoenzyme by forming an inhibitory octamer. The cryo-EM structure of RNAP-sigB holoenzyme octamer shows that the non-conserved region of sigma B is responsible for the oligomerization and the mycobacterium-specific transcription regulator RbpA releases the inhibition state. The manuscript reports a new regulatory mechanism of bacterial RNAP holoenzyme. The work overall is of high quality, and I recommend publishing after addressing below concerns.

1. Is the auto-inhibition mechanism of RNAP-sigB holoenzyme specific to *Mycobacterium tuberculosis*? How about the RNAP-sigB holoenzymes in other mycobacterial species?
2. Fig. 6C. Please label the left and right panels. I assume that the left panel shows the results without RbpA, while the right panel shows those with RbpA.
3. Video 1 is mentioned in the main text but is missing in the submitted files.
4. Page 7, paragraph 2, "the domain 4 of sigma B could not be correctly positioned for the -35 element promoter binding. This explains previous findings showing that EsigB is inactive at the -10/-35 promoters and active at the extended -10 promoters". The structure doesn't explain the promoter preference of RNAP-sigB holoenzyme, because the current octamer inhibits all types of promoters. Even if dissembled, the promoter is also not capable of initiating transcription from the extended -10 promoter as the R3.1 and R3.2 have not been properly bound RNAP core enzyme.
5. Page 13. Either Y57A or H60A abolished the octamer formation, however neither Y57A nor H60A increase the basal activity of RNAP-sigB holoenzyme in the absence of RbpA. It would be expected that disruption of the inhibitory octamer interface should increase its basal transcription activity as the sigB71omega72 mutant did.
6. The oligomerization state in solution of wild-type and derivatives of RNAP-sigB holoenzyme (Figs 5B-5E and 6A) should be confirmed by another method (for example, size-exclusion chromatography). The ratio of octamer/monomer particles in Figs 5B-5E and 6A should be quantified.
7. The authors proposed that RbpA releases the RNAP-sigB oligomerization by disrupting the octamer interface. The authors need to present the evidence showing RbpA-triggered dissociation of sigB-RNAP octamer besides the results showing pre-incubation of RbpA prevents sigB-RNAP octamer formation.
8. Some typos
 - 1) Table 2. 'Fdx -Mtb EsA' should be 'Mtb EsA-Fdx'.
 - 2) Page3, paragraph 1, line 2, 'an' should be 'and'.

Reviewer #2:

Remarks to the Author:

This manuscript describes the finding that Mtb RNAP can assemble into an octameric complex in a σ B-dependent manner. The discovery is impactful because it is the first example of a bacterial RNAP forming these higher-order complexes with biological implications. It is not well understood how σ B-RNAP and σ A can distinguish their regulons, and this finding that σ B-holo forms an immature holoenzyme without the presence of the sigma-tether RbpA provides a possible explanation. The experiments are well-executed, with commendable controls and tests using negative stain and supporting biochemical assays, and the manuscript is well-written. However, some of the data analyses (mainly statistical- see below) and more thorough explanations of the findings should be included. Once these are completed, I recommend acceptance as this is a significant finding with rigorous supporting experiments.

Major:

- Need to include 3D-FSC calculations as a plot of FSC against spatial frequency curves (global + worst and best directional FSC curves + histogram of directional FSC spatial frequencies). Because the current rendering suggests poor Euler angle distribution, these plots will allow the readers to assess the quality of the map more quantitatively. <https://3dfsc.salk.edu/>
- Is the σ_B factor occupancy of each octamer protomer unit known or evaluated? Although they were able to process and build a model of the protomer from the octamer, it would be nice to see evidence from a diagram (or just a simple mathematical calculation of the volume of the octamer EM density map relative to 8X the volumes of RNAP core + σ_B). A concern is that one (or more) of the subunits do not have a σ_B factor bound to the RNAP core (which can happen in ring-like systems for ligands like in AAA+ ATPases for regulatory purposes). Native MS would be an excellent way to assess this.
- Referring to the protomer and dimer processing: Why use a mask on those specific protomers or dimers (R1 and R5)? Does this mask pick up the other protomers that match that volume, or does it only pick up the other R1s and R5s (how would it even tell??).
- Can the authors expand on how they think RbpA expression would affect their model? My impression was that RbpA was expressed even when the cells were not under stress and, therefore, not consistent with the current model that RbpA induces the conversion of the immature Sig-B RNAP to the mature form.

Minor:

- At the beginning of the intro, the authors should cite the first papers to describe the holo- Murakami & Darst, Science 2002; Vassylyev Nature 2002 and Bae 2015 Elife.
- At the very beginning of the intro, the authors should cite Burgess et al. for RNAP subunit composite and description of sigma.
- Can the authors discuss whether they attempted to use detergents or a different surface to avoid preferred orientation issues (that the Euler angle distributions seem to suggest + the stretched-out density map views in one axis) – it also seems like there are more 'ring-face' views than 'ring rim' views
- Can the authors discuss whether they tried to purify the reconstituted E σ B holoenzyme to get rid of free RNAP core and free σ_B factor rather than doing dialysis that retained free RNAP core while getting rid of most of the σ_B factor? This could improve particle picking (thus resolution) as some side views of the octamer might not be that large in box size appearance compared to the free RNAP core. It would also just lead to more particles.
- Referring to the sigmaB NCR insertion mutants: To what extent is the insert sequence-agnostic in its oligomerization-abolishing effects? Is it just dependent on the insert length/shape?
- Fig 6C – needs clarification that the LHS graph is without RbpA and the RHS graph is with RbpA

Response letter for the manuscript: "Structural basis of the mycobacterial stress-response RNA polymerase auto-inhibition via oligomerization" by Morichaud et al.

We thank reviewers for the positive assessment of our work and for the insightful comments and suggestions to improve it. In response to the concerns raised by reviewers, we performed a set of additional experiments and data analysis. Consequently, additional panels are included in the revised main Figures 1, 5, 6 and supplementary Figures 2, 4, 8. A new supplementary Figure 5, showing more details on 3D variability analysis, was added. To improve presentation of results we moved the old supplementary Figure 9b panel to the main Figure 2d. Table1 was moved to Supplementary Table 1. Please note that to fulfill Nature Communications format guidelines we were obliged to shorten the subtitles in the "Results" section to fit the 60 character limit. We hope that our response adequately addresses the questions raised. All modifications in the revised manuscript text addressing reviewers concern are highlighted in yellow color. Below, we have included reviewer comments in *blue italic* font.

Reviewer #1 (Remarks to the Author):

Oligomerization of virus RdRP, Bacillus subtilis RNAP, eukaryotic Pol I and human mitochondrial RNAP previously have been reported as a means to regulate their transcription activity. In this study, the authors reported an auto-inhibition mechanism of Mycobacteria tuberculosis RNAP-sigB holoenzyme by forming an inhibitory octamer. The cryo-EM structure of RNAP-sigB holoenzyme octamer shows that the non-conserved region of sigma B is responsible for the oligomerization and the mycobacterium-specific transcription regulator RbpA releases the inhibition state. The manuscript reports a new regulatory mechanism of bacterial RNAP holoenzyme. The work overall is of high quality, and I recommend publishing after addressing below concerns.

1. Is the auto-inhibition mechanism of RNAP-sigB holoenzyme specific to Mycobacterium tuberculosis? How about the RNAP-sigB holoenzymes in other mycobacterial species? Considering that the octamer formation is highly sensitive to the structure of sigma NCR and likely of the beta-prime i1 insertion, it's difficult to predict. It will require testing capacity for oligomerization with purified sigmas and RNAPs from various species. We expect that sigmas displaying high homology to Mtb sigma-B, will induce octamer formation.

2. Fig. 6C. Please label the left and right panels. I assume that the left panel shows the results without RbpA, while the right panel shows those with RbpA.

We thank reviewer for noticing it, we added labels to the revised Fig 6c.

3. Video 1 is mentioned in the main text but is missing in the submitted files.

We apologize for this. We included the video file in the revised manuscript.

4. Page 7, paragraph 2, "the domain 4 of sigma B could not be correctly positioned for the -35 element promoter binding. This explains previous findings showing that EsigB is inactive at the -10/-35 promoters and active at the extended -10 promoters". The structure doesn't explain the

promoter preference of RNAP-sigB holoenzyme, because the current octamer inhibits all types of promoters. Even if disassembled, the promoter is also not capable of initiating transcription from the extended -10 promoter as the R3.1 and R3.2 have not been properly bound RNAP core enzyme.

We thank reviewer for pointing on this inconsistency. Indeed, octamer formation will inhibit ext-10 promoters as we show in Figure 6. We deleted this confusing sentence.

5. Page 13. Either Y57A or H60A abolished the octamer formation, however neither Y57A nor H60A increase the basal activity of RNAP-sigB holoenzyme in the absence of RbpA. It would be expected that disruption of the inhibitory octamer interface should increase its basal transcription activity as the sigB71omega72 mutant did.

That's right, but the transcription assay shown in Figure 6b cannot be used to evaluate an effect of octamer formation on transcription. This assay was performed at our standard reaction conditions: 100nM RNAP holoenzyme formed at 37°C for 5 min and then incubated 10 min with promoter DNA at 37°C. That is different from the condition, optimal for octamer assembly, used in experiments in Fig6c,d. In that case, 200 nM RNAP holoenzyme was incubated at +4°C overnight, and then mixed with promoter DNA at 37° (panel c) or 30°C (panel d) for different time intervals. We should note that at least 1h of incubation at 200 nM of RNAP core is required to reach a high yield of octamer.

6. The oligomerization state in solution of wild-type and derivatives of RNAP-sigB holoenzyme (Figs 5B-5E and 6A) should be confirmed by another method (for example, size-exclusion chromatography). The ratio of octamer/monomer particles in Figs 5B-5E and 6A should be quantified.

In response to this comment, we evaluated an extend of the RNAP oligomerization by the analytical size exclusion chromatography on Superose 6 Increase column. Corresponding profiles are now included as Figure 5b and Figure 6b. As can be seen, we can clearly distinguish the peak of octamer migrating as a ~3 MDa complex.

We should note that in order to detect octamer, we performed chromatography in the transcription buffer containing Mg^{2+} ions and low salt (150 mM NaCl) (please see revised Methods section for details) which is not optimal for a good peak separation. Note that core and holo RNAPs migrate mainly as a dimer in presence of Mg^{2+} ions. Furthermore, at these conditions, we observed a huge loss of protein on the column. When migration was performed under the standard conditions for RNAP purification: without Mg^{2+} and at 300 mM NaCl, which gave no loss of material and a good peak separation, we observed little or no oligomerization. Before we reported that without RbpA, sigma-A and sigma-B are weakly bound to core RNAP and dissociate under non equilibrium conditions, e.g. gel-filtration and pull-down (Hu et al., NAR 2012, 2014). Therefore, the ratio between octamer / holoenzyme / core observed in chromatography does not reflect the ratio observed at equilibrium in solution (e.g. in EM).

The fraction of RNAP molecules assembled into octamers (marked as “o-RNAP”) as a % of total RNAP molecules is now indicated in each negative stain EM panel.

7. The authors proposed that RbpA releases the RNAP-sigB oligomerization by disrupting the octamer interface. The authors need to present the evidence showing RbpA-triggered dissociation of sigB-RNAP octamer besides the results showing pre-incubation of RbpA prevents sigB-RNAP octamer formation.

In response to this comment, we added new data showing an effect of RbpA on assembly. Please see the revised Figure 6b and Figure 6e. RbpA was added to pre-formed octamer and the oligomeric state of RNAP was assessed by negative stain EM and by the size exclusion chromatography. Both methods show that addition of RbpA shifts equilibrium towards dissociation. Yet, we cannot distinguish if RbpA actively disrupts octamer or it just stabilizes a monomeric RNAP state.

8. Some typos

1) Table 2. 'Fdx -Mtb EsA' should be 'Mtb EsA-Fdx'.

2) Page 3, paragraph 1, line 2, 'an' should be 'and'.

We thank reviewer for noticing it, typos were corrected.

Reviewer #2 (Remarks to the Author):

This manuscript describes the finding that Mtb RNAP can assemble into an octameric complex in a σ_B -dependent manner. The discovery is impactful because it is the first example of a bacterial RNAP forming these higher-order complexes with biological implications. It is not well understood how σ_B -RNAP and σ_A can distinguish their regulons, and this finding that σ_B -holo forms an immature holoenzyme without the presence of the sigma-tether RbpA provides a possible explanation. The experiments are well-executed, with commendable controls and tests using negative stain and supporting biochemical assays, and the manuscript is well-written. However, some of the data analyses (mainly statistical- see below) and more thorough explanations of the findings should be included. Once these are completed, I recommend acceptance as this is a significant finding with rigorous supporting experiments.

Major:

1. Need to include 3D-FSC calculations as a plot of FSC against spatial frequency curves (global + worst and best directional FSC curves + histogram of directional FSC spatial frequencies). Because the current rendering suggests poor Euler angle distribution, these plots will allow the readers to assess the quality of the map more quantitatively. <https://3dfsc.salk.edu/>

We thank reviewer for suggestion to use 3D-FSC tool. We performed 3D-FSC calculations which are now included in supplementary Figures 2, 4, 8.

2. Is the σ_B factor occupancy of each octamer protomer unit known or evaluated? Although they were able to process and build a model of the protomer from the octamer, it would be nice to see

evidence from a diagram (or just a simple mathematical calculation of the volume of the octamer EM density map relative to 8X the volumes of RNAP core + σ^B). A concern is that one (or more) of the subunits do not have a σ^B factor bound to the RNAP core (which can happen in ring-like systems for ligands like in AAA+ ATPases for regulatory purposes). Native MS would be an excellent way to assess this.

We attempted native MS on MtbRNAP, but we were unable to find conditions at which MtbRNAP core was stable.

Our biochemical results show that sigma "locks" RNAP protomers together. Thus, dissociation of sigma should induce disassembly of the octamer and appearance of the oligomers with the number of protomers $N < 8$. If octamers missing the sigma subunit exist, they should be transient and represent a minor population which is not detectable in our sample. In support to this premise, we found a mixture of tetramers and octamers in our sample suggesting that there is an equilibrium between different oligomerization states (please see Figure S3, and text in Results p3). Please note that we used all particles corresponding to tetramers and octamers for reconstruction of the octamer D4-map to include maximum number of possible projections.

We now added volume calculations for each protomer in C1-map (revised Fig 1f) and a volume calculation for sigma-B and core in C1-map as a fraction of the volume of the atomic model (new Supplementary Fig. 5d,e). As can be seen, there is significant heterogeneity in volume in C1-map calculated using a full 115K particles dataset. When we performed "cleaning" of the dataset using 3D classification (new Supplementary Fig. 5a) we ended up with a C1-map showing little variation in the volumes of protomers. Thus, we see little variation in core and sigma volumes (compare new supplementary Fig. 5d,e).

In addition, to detect possible missing subunit, we performed 3DVA analysis in cryoSPARC on this cleaned class of octamers comprising 78k particles which were symmetry expanded relative to D4 rotation axis to 627k set (new supplementary Fig. 5d,e). 3DVA reveals only movement of the protomers relative to each other and variation in the density of the sigma C-terminal segment but no missing density of sigma domain 2 holding protomers together.

We added the following sentence in the Results section: *"Little variation in the σ^B density volume between the eight protomers in C_1 -map suggested that majority of the octamer molecules contain eight copies of the σ^B subunit (Supplementary Fig. 5d, e)."*

3. Referring to the protomer and dimer processing: Why use a mask on those specific protomers or dimers (R1 and R5)? Does this mask pick up the other protomers that match that volume, or does it only pick up the other R1s and R5s (how would it even tell??).

We used non-uniform C1-map in focused refinement. Because each protomer in C1-map displays unique map features that allowed us to distinguish them and to attribute unique names: R1 to R8. Accordingly, the R1-R5 protomers, displaying best defined density/features of RNAP core and the sigma-B domain 2 and the C-terminal segment, were chosen for masking. To clarify it, we added the following sentence in the text: *"To better characterize the $E\sigma^B$ protomer structure, we performed local refinement of the octamer C_1 -map with masked R1 and R5 protomers which displayed better defined density for σ^B "*

4. *Can the authors expand on how they think RbpA expression would affect their model? My impression was that RbpA was expressed even when the cells were not under stress and, therefore, not consistent with the current model that RbpA induces the conversion of the immature Sig-B RNAP to the mature form.*

To answer this question we should consider RbpA expression together with the expression of SigA and SigB subunits. The mRNA and protein levels of SigA and SigB in Mycobacteria were shown to be similar during exponential growth (Pettersson et al, PMID: 26445268 ; Hurst-Hess K, PMID: 31113892). RbpA, which is indeed expressed at exponential growth, was up-regulated in Mtb at starvation (9-fold) and at stress conditions (Betts et al 2002, PMID: 11929527 ; Pettersson et al.,). SigB was also up-regulated at starvation (6-fold) and stress while SigA remains unchanged or was down-regulated (Betts et al 2002, Voskuil, et al., PMID: 15207491; Aguilar-Ayala DA et al., PMID: 29247215; Pettersson et al.,). We believe these facts support our hypothesis that RbpA is required to convert SigB-RNAP into an active form. Yet, it remains only a hypothesis, because little is known on the real protein levels of these factors at different growth conditions. To better illustrate our idea we added the following sentence in the Discussion section : *"Thus, contrary to sigA, the sigB gene expression was upregulated up to 6-fold concurrently with RbpA (a 9-fold induction)"*

Minor:

1. *At the beginning of the intro, the authors should cite the first papers to describe the holo-Murakami & Darst, Science 2002; Vassylyev Nature 2002 and Bae 2015 Elife.*

References were added

2. *At the very beginning of the intro, the authors should cite Burgess et al (). for RNAP subunit composite and description of sigma.*

References were added

3. *Can the authors discuss whether they attempted to use detergents or a different surface to avoid preferred orientation issues (that the Euler angle distributions seem to suggest + the stretched-out density map views in one axis) – it also seems like there are more ‘ring-face’ views than ‘ring rim’ views*

We indeed attempted to use CHAPSO, but we observed less octamers and we stopped experiments in this direction

4. *Can the authors discuss whether they tried to purify the reconstituted EσB holoenzyme to get rid of free RNAP core and free σB factor rather than doing dialysis that retained free RNAP core while getting rid of most of the σB factor? This could improve particle picking (thus resolution) as some side views of the octamer might not be that large in box size appearance compared to the free RNAP core. It would also just lead to more particles.*

At the beginning, we attempted to purify sigB-RNAP for cryo-EM on Superose 6 size exclusion column, but without success. We succeed in analytical format (please see revised Figure 5b) but in preparative runs we observed a huge loss of material on the column and after, during the concentration using Amicon ultrafiltration units. Basically, it was impossible to achieve concentration of the sample suitable for cryo-EM. So we decided to skip purification. Without purification we observed at least ~86% of RNAP holoenzyme in octamers which we believe is a maximum we can achieve. Please see also our response to the comment 6 of reviewer 1.

5. Referring to the sigmaB NCR insertion mutants: To what extent is the insert sequence-agnostic in its oligomerization-abolishing effects? Is it just dependent on the insert length/shape?

According to our data, it can be both. Thus a large insertions in NCR tip create a steric clash with neighboring subunits. Also point mutations, which affect the residues in the interface, should compromise interactions and will destabilize octamer.

6. Fig 6C – needs clarification that the LHS graph is without RbpA and the RHS graph is with RbpA

We thank reviewer for noticing it, we added labels to the revised Fig 6C.

Reviewers' Comments:

Reviewer #1:

Remarks to the Author:

The authors have performed additional SEC experiments and quantified the single particles on the negative-staining images. The results show that the RNAP-sigma B holoenzyme forms high-order oligomers (mostly octamer) in solution and that RbpA disrupts the oligomerization, supporting the cryo-EM structure of the octameric form of the RNAP-sigma B holoenzyme. The main conclusion has been strengthened in the revised manuscript and the authors have also addressed my other concerns. Therefore, I recommended publication of the manuscript on Nature Communications.

Reviewer #2:

None

Reviewer #3:

None